# Diet Quality and Dietary Inflammatory Index Score among Women’s Cancer Survivors

**DOI:** 10.3390/ijerph19041916

**Published:** 2022-02-09

**Authors:** Sibylle Kranz, Faten Hasan, Erin Kennedy, Jamie Zoellner, Kristin A Guertin, Nitin Shivappa, James R Hébert, Roger Anderson, Wendy Cohn

**Affiliations:** 1Department of Kinesiology, University of Virginia, Charlottesville, VA 22903, USA; fh4ua@virginia.edu; 2Department of Public Health Sciences, University of Virginia, Charlottesville, VA 22903, USA; emk2fz@virginia.edu (E.K.); jz9q@virginia.edu (J.Z.); kag8c@virginia.edu (K.A.G.); ra2ee@virginia.edu (R.A.); wfc2r@virginia.edu (W.C.); 3Cancer Prevention and Control Program, Department of Epidemiology and Biostatistics, Arnold School of Public Health, University of South Carolina, Columbia, SC 29208, USA; shivappa@email.sc.edu (N.S.); jhebert@sc.edu (J.R.H.)

**Keywords:** women’s cancer survivors, cancer survivor guidelines, dietary guidance, healthy eating index, dietary inflammatory index

## Abstract

The purpose of this study was to investigate Healthy Eating Index 2015 (HEI-2015) and Energy-Adjusted Dietary Inflammatory Index (E-DII^TM^) scores in women’s cancer survivors and to examine socio-economic (SES) characteristics associated with these two diet indices. In this cross-sectional study, survivors of women’s cancers completed a demographic questionnaire and up to three 24-h dietary recalls. HEI-2015 and E-DII scores were calculated from average intakes. One-way ANOVA was used to examine the association of various demographic factors on HEI-2015 and E-DII scores. Pearson Correlation was used to calculate the correlation between the two scores. The average HEI-2015 score was 55.0 ± 13.5, lower than the national average, and average E-DII was −1.14 ± 2.24, with 29% of women having a more pro-inflammatory and 71% a more anti-inflammatory diet. Diets with higher HEI-2015 scores were associated with more anti-inflammatory diets (r = −0.67, *p* < 0.001). Those having a graduate degree (F(2,49) = 3.6, *p* = 0.03) and completing cancer treatment > 4 years ago (F(2,49) = 4.8, *p* = 0.01) had higher HEI-2015 scores. There were no associations between SES and E-DII scores. The diet quality of women’s cancer survivors is comparatively low, but many achieved an anti-inflammatory diet; a promising avenue for preventing recurrence. There is an urgent need to involve health care professionals in the guidance of women’s cancer survivors to improve diet quality and prevent cancer recurrence.

## 1. Introduction

In 2017, women’s cancers, including breast, ovarian and other cancers accounted for 41% of new cancer cases and 26% of cancer deaths in the United States (U.S.) [1]. The U.S. population has an estimated 16.9 million *cancer survivors*, a term that refers to a person with a history of cancer, beginning at diagnosis and divided into three phases: (1) time of diagnosis to the end of initial treatment; (2) transition from treatment to extended survival; and (3) long-term survival [2]. Although evidence indicates the role of inflammation in cancer risk [3,4], modifiable lifestyle factors associated with inflammation, such as diet quality [5], are understudied. Available studies on diet, cancer recurrence, and mortality in cancer survivors have shown inconsistent results [6]; however, consumption of a high-quality diet is associated with decreased mortality, specifically for those with the highest intake of vegetables and fish [7]. Breast cancer survivors consuming a Mediterranean diet appear to have reduced cancer recurrence and mortality [8].

Because consuming a high-quality diet is an important lifestyle factor to reduce morbidity and potentially prevent recurrence of cancer, the diet quality of women’s cancer survivors needs to be better understood. The American Cancer Society (ACS) and the World Cancer Research Fund/American Institute for Cancer Research (WCRF/AICR) created diet guidelines for cancer survivors for cancer prevention [9,10]. The WCRF/AICR guidelines from the Third Expert Report are part of the Continuous Update Project (CUP) and “is the world’s largest, more authoritative and up-to-date source of scientific research on cancer prevention and survivorship through diet, nutrition, physical activity, and cancer” [10]. According to WCRF, the two main parts of their work fall into determining the lifestyle factors that affect cancer and “sharing the evidence with as many people as possible”. The WCRF also notes that cancer survivors should follow the recommendations along with the guidance of their health care professionals. The ACS guidelines were based on addressing four areas of individual choices that may reduce cancer risk: weight management, physical activity, diet, and alcohol consumption. Thus, cancer prevention guidelines have shifted to a more comprehensive view by focusing on dietary patterns rather than intake levels of individual nutrients and compounds. The guidelines are based on systematic reviews and CUP provided by WCRF/AICR, as well as more recently published systematic reviews and large pooled analyses. Diet quality emerged as an important factor.

Two different measures of diet quality are the Healthy Eating Index-2015 (HEI-2015) [11] and Dietary Inflammatory Index (DII^®^), a literature-derived tool to assess the inflammatory potential of diet [12]. While the HEI measures adherence to the current intake levels of food groups used in the Dietary Guidance for Americans, depicted in the MyPlate system, the DII focusses on consumption of dietary components that modulate inflammation, including anti-oxidants. A pro-inflammatory diet (higher DII score) is associated with higher levels of inflammatory biomarkers including C-reactive protein and interleukin-6 [13,14,15,16].

To date, much knowledge of dietary intake behavior in cancer patients focuses on the treatment period and research on diet quality in cancer survivors and especially cancer survivors living in rural areas are needed [17]. The purpose of this exploratory study was to (1) describe diet quality, as measured using the HEI-2015 and DII, (2) examine demographic characteristics associated with these two diet indices, and (3) explore the correlation between the HEI-2015 and energy-adjusted DII (E-DII^TM^) in a convenience sample of rural women’s cancer survivors.

## 2. Materials and Methods

Approximately three years after a previous women’s cancer survivorship study [18], participants who indicated interest in future studies were recruited for this cross-sectional study. Questionnaires were sent to participants to collect information on demographic data (age, education, income, financial security), years since treatment, self-reported height and weight, and weight-loss goals. Rurality status was determined using 5-digit Zip-codes and rural-urban commuting area (RUCA) county characteristics [19].

After consenting to participate in the study, participants completed the questionnaire and provided up to three phone-administered 24-h recalls for two weekdays and one weekend day between April and July 2019. Data were collected by trained interviewers who called participants at their preferred day and time using Nutrient Data System for Research (NDS-R) software [version 2018, Nutrition Coordinating Center (NCC), University of Minnesota, Minneapolis, MN) [20]. Foods not included in the nutrient database were noted and estimated using the most similar available food. Inter-interviewer ratings indicated less than 5% variation. 

The Healthy Eating Index-2015 (HEI-2015), developed by the U.S Department of Agriculture (USDA), measures alignment between an individual’s diet and the 2015–2020 USDA Dietary Guidelines [11]. For women who completed more than one dietary recall, nutrients were averaged to reflect their typical diet. Total HEI-2015 scores were calculated using the SAS program developed and provided by NDS-R [21] and range from 0–100, with higher scores reflecting better adherence to intake recommendations [22,23].

DII scores were calculated using 28 parameters available from NDS-R (alcohol, vitamin B6 and B12, beta carotene, caffeine, carbohydrates, cholesterol, total fat, fiber, folic acid, iron, magnesium, MUFA, niacin, omega 3, protein, PUFA, riboflavin, saturated fat, selenium, thiamin, trans fat, vitamins A, C, D, and E, and zinc) [12]. Supplements were not included in this study. A higher DII score indicates a more pro-inflammatory diet. To control for the effect of total energy intake, E-DII scores computed per 1000 kcal of food consumed using 27 of the parameters (energy was in the denominator) [24].

Averages with standard deviations, proportions, and frequencies were calculated for each woman to describe the study sample and diet characteristics. Analysis of variance (ANOVA) was performed using RStudio to investigate the relationships among various demographic and lifestyle factors on HEI-2015 and E-DII scores [25]. Potential interactions between related demographic and lifestyle factors, such as education and income, were examined. Finally, the relationship between HEI-2015 and E-DII scores was evaluated using Pearson’s correlation coefficient [26]. The data presented in this study are available in Appendix A. 

## 3. Results

A total of 169 questionnaires were mailed to eligible participants. Two women were deceased, one declined, one questionnaire was returned in the mail, and 54 (33% of eligible) women did not respond, resulting in 111 (66%) participants completing the questionnaire. Of these, 84 (76%) expressed interest in completing dietary recalls, though one woman dropped out (cancer recurrence), one requested not to participate, and 30 were unresponsive to three call attempts. Thus, 52 women (47%) completed at least one diet recall (*n* = 35 provided three, *n* = 7 two, and *n* = 10 one day of intake data) for a total of 129 dietary recalls included in these analyses.

On average, participants were 65 ± 12 years old; 94% White; 96% non-Hispanic; 71% married; 47% with graduate education; all had health insurance coverage; 60% lived in urban residences. Breast cancer was the most common cancer type (56%), followed by endometrial/uterine (29%) and ovarian (10%). Prior treatment received for the initial cancer diagnosis included surgery only (25%), surgery combined with radiation (25%) or chemotherapy (8%), or all three treatments combined (21%). About two-thirds reported having at least one other pre-existing condition (in addition to cancer), with nearly one-fifth (19%) reporting having at least 3 pre-existing conditions. These include high cholesterol (33%), high blood pressure (31%), arthritis (25%), depression (15%), anxiety (13%), diabetes (8%), neuropathy (4%), and other pre-existing conditions (15%). At the time when the dietary recalls were conducted, average time since completion of active cancer treatment was 4.25 years.

Results showed that, on average, participants did not meet the Dietary Guidelines for Americans for fruits, vegetables, seafood and plant protein, and refined carbohydrates (Table 1). Average macronutrient distribution of calories in women was 45.0% carbs, 16.8% protein, 35.7% fats (12.1% saturated fats, 12.5% monounsaturated fats, and 8.2% polyunsaturated fats,). On average, women consumed 1711.8 kilocalories, 197.9 g carbohydrates (18.8 g dietary fiber), 70.2 g protein, and 70.4 g fats (23.7 g saturated fats, 24.8 g monounsaturated fats, 16.3 g polyunsaturated fats).

The average HEI-2015 total score was 55.0 ± 13.5 and ranged from 29.7 to 84.6 points (median 56.0) and the average E-DII score was −1.14 ± 2.24, ranging from −5.66 to 3.22 (median −1.24).

Women with a graduate degree and women who completed their last active treatment more than four years prior to this study had higher HEI-2015 (*p* = 0.009 and *p* = 0.01, respectively; Table 2). None of the socio-economic characteristics included in this study were significantly associated with E-DII scores. There was a significant inverse correlation between HEI-2015 and E-DII, such that higher HEI-2015 scores correlated with lower E-DII (Figure 1, r= −0.67, *p* < 0.001).

## 4. Discussion

This study was conducted to assess the diet quality and, consequently, the potential risk for cancer recurrence and morbidity in women’s cancer survivors. Surprising results included that the average HEI-2015 total score was only 55.0 of possible 100 points and had a wide range (29.7 to 84.6 points), which indicates a lower diet quality than the United States national average of 59 points [27]. These findings are in line with others reporting low diet quality in cancer survivors [28,29].

Participants’ E-DII scores also indicate reason for concern, as 29% of women’s cancer survivors had a pro-inflammatory diet. Inflammation is the substrate underlying many mechanisms contributing to several types of cancer [9,10] and growing evidence supports the role of diet in inflammatory pathways in both carcinogenesis and disease recurrence. In addition, higher DII/E-DII scores are correlated with increased inflammatory markers [24,30], increased risk of cardiovascular disease and metabolic syndrome [24,31], and 75% increased mortality for breast, colorectal and lung cancers [32,33], but not endometrial cancer, except in those who are very obese [31]. Thus, women’s cancer survivors might considerably reduce the risk of cancer recurrence, if they could achieve a more anti-inflammatory diet.

Considering the impact of diet on cancer risk, it appears that women’s cancer survivors do not receive consistently appropriate information on how to achieve a higher quality diet to support health and prevent the future development of cancer or cancer relapse. Thus, there is a need for better dietary guidance for cancer patients. Effective communication of diet quality goals and offers of support for cancer survivors may be needed to achieve these goals.

Only two socio-economic characteristics, education and years since treatment, were associated with HEI-2015 scores, yet these characteristics were not significantly associated with E-DII scores. No interactions were found between related characteristics. Although age, education, BMI, and income are established correlates of diet quality in the general population [34,35,36] and amongst cancer survivors [28], we did not observe that relationship. However, the participants in this study were considerably younger (65.0 ± 12.0 years vs. 57.6 ± 10.1) and most were White and financially secure compared to the population studied by Springfield et al. [28]. Furthermore, we included other women’s cancers (in addition to breast) that tend to have poorer prognoses (e.g., ovarian), compared to a study population that included only breast cancer survivors.

As expected, there was a significant inverse correlation between HEI-2015 and E-DII, which is consistent with other research [37]. This is a significant finding, especially in women’s cancer survivors, who may be at risk for recurrence of cancer. As HEI-2015 scores increased, DII scores decreased, indicating better diet quality and lower inflammatory diet patterns. Most importantly, this result shows that guidance and support to achieve and maintain high diet quality and anti-inflammatory diet patterns is critical. Currently, dietary guidelines for cancer survivors are to follow those for cancer prevention [9,10]. In general, the existing guidelines for cancer survivors do not differ greatly from those for the general population, despite potentially different nutritional needs. The WCRF/AICR Recommendations for Cancer Survivors are less specific than both ACS and HEI-2015 guidelines. The WCRF/AICR guidelines state to “be healthy weight, be physically active, eat more whole grains, vegetables, fruits and legumes (such as beans), avoid sugary drinks, limit consumption of “fast foods” and other processed foods high in fat, starches, or sugars, limit red meat consumption, and avoid processed meats and alcohol.”. Furthermore, guidelines state that “Foods containing fiber and soy decrease mortality risk before and ≥12 months after diagnosis, while total fat and saturated fat increase mortality risk before diagnosis.” These guidelines overlap with the HEI-2015 guideline’s recommendation to moderate added sugar intake to ≤6% of total energy, moderate saturated fat intake to ≤8% of total energy and consume ≥1.5 oz eq of whole grains per 1000 kcal. The ACS guidelines overlap with HEI-2015 guidelines in 10 of the 13 components, though only the recommendation to “Eat at least 2.5 cups of vegetables and fruits each day” provides a quantified guideline. In the remaining seven areas of overlap with the HEI-2015 guidelines, the ACS guidelines suggest to “limit…”, “avoid…”, “minimize…”, and “choose…instead of…”. The HEI-2015 components not addressed by the ACS guidelines are recommendations on dairy consumption, total protein foods, and sodium. Furthermore, the ACS guidelines suggest “No more than 2 alcoholic drinks per day for men or 1 drink per day for women”, while the HEI-2015 guidelines do not address alcohol. However, breast cancer survivors recently received a more tailored set of recommendations by the ACS based on a systematic literature review conducted in 2015. These guidelines emphasize the importance of consuming a diet high in vegetables, fruits, whole grains, and legumes, while limiting saturated fats and alcohol. Specifically, the guidelines state that “dietary changes sufficient to result in weight loss may be needed to favorably impact breast cancer recurrence and prognosis” [38].

Dietary guidelines are particularly important with respect to body weight management due to the established link between obesity and other comorbidities and cancer survival and recurrence. Specifically, meta-analyses have uncovered associations between BMI status at endometrial cancer diagnosis and cancer recurrence and all-cause mortality [39]. While BMI was not associated with cancer-specific mortality in this study, a relationship was observed between visceral fat and cancer-specific mortality in two of the studies included in the meta-analysis [40,41]. This is extraordinarily important because the effect of obesity is mediated nearly exclusively by adipose tissue. So, while there is a strong positive correlation between obesity and adiposity, it is the latter that is the most important in terms of modulating inflammation [42,43]. A meta-analysis by Secord et al. [44] found that endometrial cancer patients with BMI > 40 had an odds ratio for all-cause mortality of 1.66 compared to women with BMI < 25. However, the study did not distinguish between all-cause mortality and cancer-specific mortality in this study, and it is well established that obesity alone increases mortality risk [45]. Nonetheless, obesity in endometrial cancer survivors is also associated with poorer quality of life, particularly regarding physical and social functioning [46,47]. These obesity-based results are consistent with the adiposity result in the meta-analyses [40,41] in that it is clear that individuals who are morbidly obese carry massive amounts of adipose tissue. Interestingly, there is a more well-established relationship between BMI and cancer recurrence for breast cancer patients. Obese breast cancer patients, relative to normal-weight patients, had increased risk of cancer recurrence and cancer-specific mortality (12.2% and 6.9%, respectively), but no association between weight status and all-cause mortality [48,49]. Additionally, weight gain after breast cancer diagnosis, especially in the form of adipose tissue, was also associated with increased all-cause mortality and recurrence [50], particularly if individuals gain greater than 10% of their body weight [51,52]. Unfortunately, weight gain is very common following cancer diagnosis, independent of the treatment used, particularly in those who already have an elevated BMI [53]. Furthermore, weight gain is generally even greater in those who receive adjuvant chemotherapy [53,54], and leaves cancer survivors at an increased risk of cancer recurrence and development of comorbid conditions [55].

Other conditions, such as diabetes, dyslipidemia, and having three or more metabolic comorbidities were all associated with increased risk of developing recurrent breast cancer [56]. Similarly, a Danish breast cancer study found that having any comorbidity was associated with increased all-cause mortality, while only certain comorbidities such as dementia, chronic pulmonary disease, peripheral vascular disease, and liver and renal diseases were associated with increased risk of cancer-specific mortality. This risk was even greater if the comorbidity was diagnosed within 5 years of cancer diagnosis, as opposed to more than 5 years since diagnosis [57]. A similar study in Shanghai found that diabetes and history of stroke were associated with an increased risk of all-cause and non-breast cancer mortality, respectively [58].

Thus, these associations between BMI, comorbidities, and cancer outcomes highlight the importance of lifestyle interventions in cancer patients and cancer survivors. While dietary interventions have shown promising results for improving anti-inflammatory nature of the diets of cancer survivors, measures are needed to extend this information to the general population of cancer survivors [59].

More data describing diet quality among cancer survivors, including rural populations, are needed to understand biologic mechanisms that may underlie the associations between diet and survivorship outcomes, tailor guidelines, and design and implement dietary interventions. More specific guidelines would allow healthcare professionals to provide specific, evidence-based oncology nutrition services, such as nutrition education, counseling, and medical nutrition therapy (MNT) [60,61].

Limitations of this study include the use of dietary self-reports, which rely on memory and have the potential for under-reporting of foods—a recognized caveat in nutrition research [62,63]. This study also did not measure gastrointestinal symptoms common during cancer treatment, such as nausea, vomiting, dry-mouth, constipation, changes in taste and loss of appetite, all of which may impact diet [64,65]; though it must be noted that all women included in this study had completed treatment at least 3 years prior to the dietary recalls. Future studies including the long-term effects of cancer treatment on GI function might elucidate the potential for long-term adverse effects affecting the potential for consuming a high-quality diet. Additionally, representativeness of our study sample and generalizability to other women’s cancer survivors should be considered with interpreting and applying these findings. Our study included a relatively small sample of largely White and highly educated women. Additionally, self-selection into the study may reflect underlying sample biases, such as over representation of those with an interest in diet.

Despite these limitations, this exploratory study addresses an important gap in cancer survivorship research. Our study included a high percentage of women living in rural areas (40%), an understudied population in the field. Future studies with larger and more diverse samples (i.e., race/ethnicity, SES, rural/urban, different cancer types) are needed to better understand: (1) diet quality among women’s cancer survivors; (2) predictors of diet quality; and (3) relationships among diet quality and survival time, quality of life, and inflammatory biomarkers (CRP, IL-6, etc.).

## 5. Conclusions

Overall, our study indicates the need to promote improvements in diet quality among women’s cancer survivors. Consuming a diet that meets the Dietary Guidelines for Americans will also likely result in a more anti-inflammatory diet, potentially helping to prevent cancer development or recurrence. Healthcare professionals may be critical partners in efforts to achieve higher HEI-2015 and lower E-DII scores to help protect from cancer recurrences. The need for more specific dietary guidance for women’s cancer survivors should be further explored.

## Figures and Tables

**Figure 1 ijerph-19-01916-f001:**
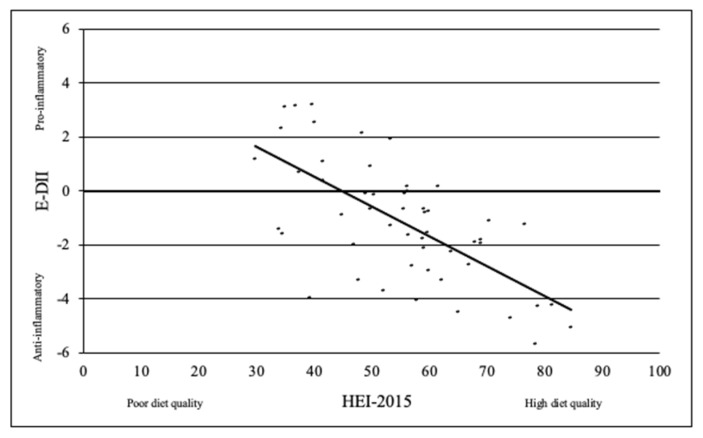
Correlation between Energy-Adjusted Dietary Inflammatory Index (E-DII) scores and Healthy Eating Index (HEI-2015) scores in women’s cancer survivors (n = 52).

**Table 1 ijerph-19-01916-t001:** Mean HEI-2015 total and component scores among women’s cancer survivors (n = 52).

	Maximum Score	Mean	SD	Range
**Total Healthy eating index (HEI-2015)**	100.0	55.0	13.5	29.7–84.6
**Total Vegetables**	5.0	3.3	1.4	0.3–5.0
**Greens and Beans**	5.0	2.0	1.8	0.0–5.0
**Total Fruit**	5.0	2.3	1.6	0.0–5.0
**Whole Fruit**	5.0	2.7	1.9	0.0–5.0
**Whole Grains**	10.0	3.8	3.0	0.0–10.0
**Dairy**	10.0	5.5	2.7	0.0–10.0
**Total Protein Foods**	10.0	4.4	0.9	0.7–5.0
**Seafood and Plant Protein**	5.0	2.6	1.9	0.0–5.0
**Fatty Acid Ratio**	10.0	4.6	3.1	0.0–10.0
**Sodium**	10.0	4.7	2.9	0.0–9.4
**Refined Grains**	10.0	6.6	2.7	0.0–10.0
**Added Sugars**	10.0	7.5	2.4	0.0–10.0
**Saturated Fats**	10.0	4.9	2.8	0.0–10.0

**Table 2 ijerph-19-01916-t002:** Diet quality as measured by the Healthy Eating Index (HEI-2015) and Energy-adjusted Dietary Inflammatory Index (E-DII) scores among women’s cancer survivors, and differences by demographic characteristics (n 52).

			HEI Total Score ^a^			E-DII ^a^		
	n	%	Mean	SD	F-Value	Mean	SD	F-Value
**Overall Scores**	52	100	55.0	13.5		−1.14	2.24	
**Age**								
**36–55**	9	17	56.5		F(3,48) = 0.91	−1.17		F(3,48) = 0.64
**56–64**	16	31	57.7			−1.76		
**65–74**	15	29	50.2			−0.72		
**75+**	12	23	56.3			−0.84		
**Education ^b^**								
**HS-4yr college**	31	60	51.1		F(1,50) = 7.4 *	−0.81		F(1,50) = 1.73
**Masters-PhD**	21	40	60.9			−1.64		
**Income**								
**$0–35 k**	14	27	55.2		F(2,45) = 1.73	−1.50		F(2,45) = 1.49
**$35–75 k**	14	27	49.3			−0.23		
**$75 k+**	20	38	58.2			−1.45		
**Not reported**	4	8						
**Financial security**								
**Living comfortably**	30	58	56.0		F(1,50) = 0.35	−1.07		F(1,50) = 0.07
**Other (getting by on present income, finding it difficult on present income, and finding it very difficult on present income)**	22	42	53.7			−1.24		
**Rurality**								
**Urban**	31	60	56.1		F(1,50) = 0.49	−1.39		F(1,50) = 0.95
**Rural**	21	40	53.4			−0.78		
**Years since treatment ^c^**								
**3–4 years**	20	38	48.2		F(2,49) = 4.8 **	−0.44		F(2,49) = 1.75
**4–5 years**	25	48	59.2			−1.67		
**>5 years**	7	13	59.9			−1.29		
**Body mass index (BMI)**								
**<24**	12	23	57.0		F(3,48) = 1.11	−0.84		F(3,48) = 1.96
**25 to <30**	18	35	58.1			−2.11		
**30 to <40**	16	31	52.7			−0.37		
**>40**	6	12	48.0			−0.93		
**Weight-loss goals**								
**Lose weight**	39	75	56.1		F(1,50) = 0.95	−1.26		F(1,50) = 0.41
**Maintain/gain**	13	25	51.9			−0.80		

HEI-2015, Healthy Eating Index-2015, E-DII, Energy-adjusted (kcal) DII (E-DII) score. ^a^ Calculated from 24-h dietary recalls of food items (does not include dietary supplements). ^b^ HEI-2015 for “>graduate” is significantly greater than “HS-college”. ^c^ HEI-2015 for “4–5 years” and “>5 years” both significantly greater than “<4 years”. * *p* < 0.05, ** *p* < 0.01.

## Data Availability

Please contact the authors to request the data supporting this publication is available from the authors.

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
