# Peer review of "Diet Quality and Dietary Inflammatory Index Score among Women’s Cancer Survivors"

_ijerph, 2022, doi:10.3390/ijerph19041916_

Round 1
Reviewer 1 Report
Kranz and colleagues followed female cancer survivors' quality of food intake. The study is clearly presented and the results are carefully discussed. Especially the limitation of the study (n size, education, white-females) is well described. The conclusion section is highly appreciated, a closer follow-up to achieve higher HEI-2015 scores is necessary, this study is a foundation for these efforts.
There are only a few minor changes that can be addressed:
1. Please give the standard deviation for the average HEI-2015 (line 144 citation 25) and state region assessed.
2. line 173/174: elaborate which recommendations
3. Please discuss BMI´s impact on cancer survival/recurrence.
4. Please discuss co-morbidities (e.g. inflammatory diseases) and the impact on cancer recurrence/occurrence.
Author Response
Response to reviewers. We indicate our response in red text and used track changes in the manuscript to highlight our changes in the text.
Kranz and colleagues followed female cancer survivors' quality of food intake. The study is clearly presented and the results are carefully discussed. Especially the limitation of the study (n size, education, white-females) is well described. The conclusion section is highly appreciated, a closer follow-up to achieve higher HEI-2015 scores is necessary, this study is a foundation for these efforts.
We thank the reviewer for the positive feedback and for taking the time to review our manuscript.
Point 1: Please give the standard deviation for the average HEI-2015 (line 144 citation 25) and state region assessed.
Response 1: We thank the reviewer for this suggestion. We have included the state region assessed, but the standard deviation for the average HEI-2015 is not reported in this data set from the National Health and Nutrition Examination Survey (NHANES).
Point 2. line 173/174: elaborate which recommendations
Response 2: We have included more information on the recommendations specific to our population.
Point 3. Please discuss BMI´s impact on cancer survival/recurrence.
Response 3: We added this information in the discussion section of our manuscript (lines 366-394).
Point 4. Please discuss co-morbidities (e.g. inflammatory diseases) and the impact on cancer recurrence/occurrence.
Response 4: We appreciate this suggestion and have included this information in the discussion section of our manuscript (lines 395-404).

Reviewer 2 Report
The original work was presented to me for review. I consider the authors' work important from the cognitive point of view. There is a lack of good-quality works about quality of diet patients with cancer.
However, I have a couple of considerations listed below:
Lines 66, 169 and 188: double space
Line 100-104: In my opinion, this information and the table should be presented below, following the anthropometric characteristics of the group. The results will be clearer.
The results do not include information on the energy and nutritional supply of the diet, the amount of protein consumed per kilograms of body weight, the type of fats consumed, the amount of dietary fiber in the diet, which significantly reduces the quality of work.
In addition, have the researchers considered gastrointestinal problems (diarrhea, constipation, vomiting) that may occur during cancer treatment and their possible impact on diet?
The "Discussion" section is written correctly, but in my opinion the number of cited studies is insufficient - it is worth describing additional observations of other authors.
Noteworthy is an accurate description of the limitations of the research, and the prospects for further work that should be developed on a given topic.
Author Response
Response to reviewers. We indicate our response in red text and used track changes in the manuscript to highlight our changes in the text.
The original work was presented to me for review. I consider the authors' work important from the cognitive point of view. There is a lack of good-quality works about quality of diet patients with cancer.
We thank the reviewer for reviewing our manuscript.
Point 1: Lines 66, 169 and 188: double space
Response 1: The double spaces have been corrected
Point 2: Line 100-104: In my opinion, this information and the table should be presented below, following the anthropometric characteristics of the group. The results will be clearer.
Response 2: We thank the reviewer for this suggestion. We have rearranged the results accordingly.
Point 3: The results do not include information on the energy and nutritional supply of the diet, the amount of protein consumed per kilograms of body weight, the type of fats consumed, the amount of dietary fiber in the diet, which significantly reduces the quality of work.
Response 3: As suggested, we included information about energy and nutritional supply of the diet, the type of fats consumed, and the type of dietary fiber of the diet (lines 206-210). Many participants did not report their body weight and we were not able to calculated grams of protein per kg body weight for the population.
Point 4: In addition, have the researchers considered gastrointestinal problems (diarrhea, constipation, vomiting) that may occur during cancer treatment and their possible impact on diet?
Response 4: We thank the reviewer for this suggestion as it indicates that our manuscript was not clearly worded. We reworded the text to clarify that we only invited participants who had finished treatment at least 3 years ago (cancer survivors) and our questionnaire did not include retrospective reports of GI problems encountered during the active treatment phase. We did not collect data on gastrointestinal problems in the post-treatment phase but we agree that future studies should include this component because cancer treatment might have a possible impact on GI problems and diet long after treatment has been concluded.
Point 5: The "Discussion" section is written correctly, but in my opinion the number of cited studies is insufficient - it is worth describing additional observations of other authors.
Response 5: We appreciate this feedback. We added more citations to support our discussion to provide additional observations from other authors that have conducted similar studies.
Point 6: Noteworthy is an accurate description of the limitations of the research, and the prospects for further work that should be developed on a given topic.
Response 6: We thank the reviewer for this comment and hope that our research highlights the need for further work in larger and more diverse populations.

Round 2
Reviewer 2 Report
I would like to thank the authors for the changes.